# Using clinical cascades to measure health facilities' obstetric emergency readiness: testing the cascade model using cross-sectional facility data in East Africa

Bridget Whaley [1], Elizabeth Butrick [2], Jessica M Sales [1], Anthony Wanyoro [3], Peter Waiswa [4], Dilys Walker [2,5], John N Cranmer [6]

For numbered affiliations see end of article.

**Correspondence to**
Bridget Whaley;
whaley.bridget@gmail.com

## ABSTRACT

**Objectives** Globally, hundreds of women die daily from preventable pregnancy-related causes, with the greatest burden in sub-Saharan Africa. Five key emergencies—bleeding, infections, high blood pressure, delivery complications and unsafe abortions—account for nearly 75% of these obstetric deaths. Skilled clinicians with strategic supplies could prevent most deaths. In this study, we (1) measured facility readiness to manage common obstetric emergencies using the clinical cascades and signal function tracers; (2) compared these readiness estimates by facility characteristics; and (3) measured cascading drop-offs in resources.

**Design** A facility-based cross-sectional analysis of resources for common obstetric emergencies.

**Setting** Data were collected in 2016 from 23 hospitals (10 designated comprehensive emergency obstetric care (CEmOC) facilities) in Migori County, western Kenya, and Busoga Region, eastern Uganda, in the Preterm Birth Initiative study in East Africa. Baseline data were used to estimate a facility's readiness to manage common obstetric emergencies using signal function tracers and the clinical cascade model. We compared emergency readiness using the proportion of facilities with tracers (signal functions) and the proportion with resources for identifying and treating the emergency (cascade stages 1 and 2).

**Results** The signal functions overestimated practical emergency readiness by 23 percentage points across five emergencies. Only 42% of CEmOC-designated facilities could perform basic emergency obstetric care. Across the three stages of care (*identify*, *treat* and *monitor-modify*) for five emergencies, there was a 28% pooled mean drop-off in readiness. Across emergencies, the largest drop-off occurred in the treatment stage. Patterns of drop-off remained largely consistent across facility characteristics.

**Conclusions** Accurate measurement of obstetric emergency readiness is a prerequisite for strengthening facilities' capacity to manage common emergencies. The cascades offer stepwise, emergency-specific readiness estimates designed to guide targeted maternal survival policies and programmes.

**Trial registration number** NCT03112018.

## Strengths and limitations of this study

► Measuring emergency obstetric readiness according to the clinical cascades provides a more nuanced picture of where readiness is lost for emergency care; cascade estimates support targeted strategies for closing gaps in critical emergency obstetric supplies.

► To the authors' knowledge, this is the first study to compare signal functions and clinical cascade estimates of obstetric emergency readiness across countries.

► This study expands previous scholarship by measuring emergency readiness by level of care (caesarean section capability), facility ownership (government vs private hospital) and country (Kenya vs Uganda).

► We did not have tracer item data to estimate readiness for assisted vaginal deliveries—one of six signal functions and clinical cascades.

► Resource availability alone does not comprehensively describe emergency readiness and a clinician's ability to use the resources and perform targeted interventions is a critical aspect; however, skill and knowledge are not easily measured and not included in either the signal function tracers or clinical cascade methodology.

## INTRODUCTION

In 2017, more than 800 women died each day from preventable pregnancy-related and childbirth-related causes, resulting in nearly 300 000 annual deaths.[1 2] A few major obstetric emergencies account for nearly 75% of these deaths: severe bleeding (haemorrhage), infections (maternal sepsis), high blood pressure (pre-eclampsia and eclampsia), delivery complications (prolonged or obstructed labour) and unsafe abortions.[3] Common obstetric emergency deaths can be effectively prevented when skilled healthcare providers have the supplies necessary to identify and

manage emergencies and can transfer patients to higher levels of care when an emergency exceeds the facility's scope of care.[3] Labour-related complications require immediate and easy access to high-quality emergency obstetric care. Basic emergency obstetric care (BEmOC) facilities are the first level of emergency management for the most common causes of maternal death. BEmOC requires essential supplies, durable equipment and emergency-specific drugs. When obstetric emergencies are beyond the capacity of BEmOC facilities, patients can be referred to comprehensive emergency obstetric care (CEmOC) facilities. CEmOC includes all BEmOC functions and adds resources for blood transfusion and surgery (such as caesarean section; C-section).[3] Mobilising and dispensing the clinical resources required to manage basic obstetric emergencies may be a critical step for reducing the persistently elevated maternal mortality ratios (MMRs) in sub-Saharan African contexts such as Kenya and Uganda. Therefore, accurate measurement of a facility's readiness to manage the common emergencies that drive maternal deaths is urgently needed.

The WHO first identified the resources necessary to manage common obstetric emergencies in the 1990s.[4] This recommended approach evolved into the 'signal functions', consisting of six clinical actions (three medical treatments and three manual procedures) for managing common obstetric emergencies. BEmOC includes three medical treatments and three manual procedures. The three medical treatments include administering parenteral (1) antibiotics, (2) uterotonics, and (3) anticonvulsants/antihypertensives. The three manual procedures include (4) manually removing retained placentas, (5) removing retained products of conception, and (6) performing assisted vaginal deliveries (AVD).[5] CEmOC extends BEmOC care by adding two additional actions: performing (7) obstetric surgeries (such as C-section) and (8) blood transfusions.[5] Thus, all designated CEmOC facilities should be prepared to perform all BEmOC signal functions, in addition to offering surgery and blood transfusion. In the signal function model, specific items (ie, 'tracers') that are considered most critical for performing the clinical actions are used as proxies to measure a facility's capacity to manage obstetric emergencies.[5–11]

Signal function estimates of emergency readiness have been the dominant method for assessing obstetric readiness at facilities worldwide.[12–15] BEmOC signal function indicators have been used to estimate a facility's overall BEmOC emergency readiness using the percent of tracer items present at the facility.[15] WHO's Service Readiness Index (SRI) defines a facility's obstetric emergency readiness using the six clinical actions that define BEmOC.[5] In the SRI methodology, the average number of tracer items present on the day of the facility assessment determines a facility's overall obstetric emergency readiness.[14 15] This method uses the signal functions framework. Traditionally, signal function readiness estimates are reported as a single indicator, the proportion of facilities with the tracer items for all medical treatments and manual procedures.

In recent years, scientists and practitioners have increasingly called for revised approaches to address weaknesses in the signal functions.[7 9 16–21] One particular weakness is the signal function's inability to predict readiness for *specific* emergencies.[16 18] Further, this approach does not identify strategic resources for managing multiple emergencies nor suggest indicators for system-wide emergency readiness.[21] Also, the signal functions do not distinguish between the resources required to first *identify* the emergency and then the consumable supplies and durable goods required to deliver targeted treatment/intervention for the emergency.[21] Thus, to further reduce delivery-related mortality, a more robust set of measurable indicators that are emergency specific and relevant for multiple levels of the health system are needed.[21] The clinical cascade model is designed to fill these gaps.

The cascade model was designed to measure BEmOC readiness at global emergency obstetric facilities. It is informed by three models—the original signal functions, the systemic capacity hierarchy of needs and the clinical care continuum (spectrum of engagement).[22 23] The cascades were designed as a clinically oriented, population-relevant family of indicators. The core indicators measure facility readiness to manage the six common obstetric emergencies by reporting the percentage of facilities with emergency resources to identify, treat and monitor-modify therapy.[21–23] Operationally, the cascades define emergency readiness as the proportion of facilities that have the resources (including the drugs, supplies and equipment) to *identify* an emergency and *treat* it. A facility's ability to *monitor and modify* the initial treatment as clinically indicated based on patient response was not a foundational measure of readiness for the presenting emergency. Rather, it has been proposed as an indicator of care quality.[21]

Unlike the wealth of research available on signal functions, the clinical cascades have been tested in fewer geographic locations. There are limited studies comparing this emerging metric with the signal functions as methods to assess facilities' readiness to handle basic obstetric emergencies. To date, one published study compared estimates of obstetric readiness based on signal functions and clinical cascades. Two additional studies were conducted in Guatemala and Ethiopia and are pending publication. The published study from BEmOC-designated facilities in Kakamega County, Kenya, found a 55% overestimate of practical emergency management readiness using the signal function tracers compared with the clinical cascades.[21] Moreover, a 33% pooled mean drop-off in readiness across three stages for all cascades (emergencies) was identified.[21] A similar pattern of emergency readiness loss also occurred for newborn emergencies in a published study from Kenya and Uganda. In those countries, a pooled mean 30% drop-off in readiness was measured across the three stages of care for all newborn emergencies.[24] This study, however, did not compare clinical cascade-based estimates of readiness against signal function estimates since no standard signal

function indicators for newborns exist.[9] The findings from these two published clinical cascade studies warrant additional investigation to determine the relevance and transferability of the obstetric clinical cascade model for other global contexts, health systems and levels of facility-based care. A more comprehensive analysis of a facility's readiness to handle obstetric emergencies, such as the cascade analysis, may be critical to guide emergency-specific or supply-specific interventions to close gaps in the availability of emergency supplies that are critical to maternal survival.

## Aims

Building off the wealth of research on the signal functions and formative research on clinical cascades, this cross-sectional facility-based study was designed to assess readiness for pregnancy-related emergencies at designated BEmOC and CEmOC facilities in Migori County, Kenya, and the Busoga Region of Uganda. The facility data used in this analysis were originally collected as part of a pair-matched, cluster randomised controlled trial evaluating a package of facility-based interventions to improve care for preterm infants.[25] This nested analysis was designed to: (1) estimate facility readiness to manage common obstetric emergencies using clinical cascades and signal functions; (2) compare these estimates based on differences in facility characteristics (eg, designated level of care, facility ownership and country); and (3) test if the cascading loss of emergency obstetric resources first identified in Kenya was also present in other global settings. The existing studies on the cascade model suggest that using signal functions to assess maternal emergency readiness may convey a false sense of security by indicating a higher level of emergency preparedness than practically exists in facilities.[21 24]

## METHODS

### Study design and data collection

We conducted a cross-sectional analysis of facility data collected in 2016–2019 by the Preterm Birth Initiative (PTBi) Kenya and Uganda Implementation Research Collaborative.[25] Data from preintervention baseline facility assessments at 23 facilities in the PTBi study were used to create the signal function and clinical cascade estimates of emergency obstetric readiness.[25] Data from all 17 PTBi facilities in Migori County (western Kenya) and six in Busoga Region (eastern Uganda) were used. We used the baseline data (2016) to capture facility readiness prior to any PTBi interventions.

Research assistants used standardised forms to visually identify emergency resources during the on-site physical inventory of resources. They captured data about facility characteristics, obstetric drugs, consumable supplies, durable goods and the presence of emergency guidelines and protocols. Researchers recorded both the presence/absence of the item and its location (ie, unit). In this analysis, we used a resource's presence or absence at the facility level to estimate facility-level readiness regardless of the unit in which the items were located.

## Emergency readiness

### Signal functions

Traditionally, signal function readiness estimates are reported as a single indicator, the proportion of facilities with the tracer items for all medical treatments and manual procedures.[21] However, for the purposes of our analysis, we reported additional indicators from the clinical cascades: (1) total readiness estimates across all SRI/signal functions; (2) readiness estimates by type of signal function (medical treatment vs manual procedure); and (3) individual readiness estimates for each signal function. This approach allowed for a direct comparison to clinical cascade estimates.

### Clinical cascades

Given that the signal functions do not have quality indicators for monitoring and modifying the primary treatment based on a patient's clinical response (stage 3 of the clinical cascades), we defined mean clinical cascade readiness using stages 1 and 2 of the clinical cascades (*identify* and *treat*). Although readiness at stage 3 of the clinical cascade of care is not reported in comparative analyses, it is included in the analyses that examine readiness along the three stages of clinical care (*identify, treat, monitor-modify*). The cascading drop-off in readiness across these three stages is an independent indicator that is not used to directly compare signal function and clinical cascade estimates. Rather, it has been proposed as a proxy indicator for a facility's readiness to monitor the initial therapy and modify that therapy based on patient response. This proposed quality of care indicator only uses commodities (eg, protocols, medications, supplies) and does not measure clinician skill. We measured clinical cascade estimates of readiness at the individual resource level (calculating percentages of facilities with each individual resource). This approach allowed us to demonstrate precisely where in the clinical cascade readiness drops off. We estimated overall emergency readiness across sites as the percentage of facilities with all resources necessary to complete the *identify* and *treat* stages of care. Of note, readiness at each of the three stages of clinical care (*identify* the emergency, *treat* it and *monitor/modify* therapy) is cumulative. For example, readiness at stage 2 is calculated from the percentage of facilities with all resources required to first identify the emergency (stage 1).

## Operational definitions

In the absence of certain variables, we used proxies for the missing tracer items to not unduly penalise a facility's readiness estimate using the SRI/signal function tracers—this was especially true for consumable supplies and durable goods, such as refrigeration (electrical power as proxy for refrigeration) and light source (electrical power or operational flashlight as proxy for a light source). Moreover,

some tracer items, namely drugs, are not concretely defined in the SRI/signal function model. When drugs were not explicitly defined by the SRI/signal functions, we referred to WHO's first-line recommendations for obstetric care to create operational definitions.[5 26 27] For instance, in the SRI/signal functions, antibiotics are broadly defined as 'parenteral antibiotics'. To transform this broad drug category into an operational indicator, we used WHO's three-step sequence of obstetric antibiotic therapy escalation to define readiness (ampicillin, gentamicin and metronidazole).[5 26 27] In other instances, data on both tracers and potential proxies were missing. This was true for most items from the AVD cascade. Consequently, we were unable to measure AVD readiness using the SRI/signal functions or the clinical cascades. Online supplemental table S1 provides a list of the emergency resources used to define the SRI/signal functions and clinical cascades.

### Analysis

We first described emergency resource availability for individual resources using percentages. In describing the availability of emergency resources, we reported percentages and frequencies for categorical variables. For continuous variables, we reported the medians with IQRs because they were not normally distributed (eg, delivery volume).

Aggregate readiness across multiple emergencies or multiple stages of care was reported as the overall pooled mean. Drop-offs in readiness (ie, the percentage of facilities prepared to handle the five emergencies) between each stage of care and across each emergency cascade were reported as percentages with the SD. We reported estimates of overall emergency readiness using the mean for several reasons: (1) the SRI/signal function method reports means, and we wanted this study's findings to be benchmarked against globally reported indicators; (2) means are most frequently used in the published literature; and (3) with few observations, the median is unlikely to effectively represent central tendency.

For the purposes of comparing the clinical cascades and SRI/signal functions, we reported: (1) mean overall (pooled) readiness across all five emergencies for the signal functions and clinical cascades; (2) mean overall (pooled) readiness by type of signal function (medical treatment vs manual procedure); (3) mean overall (pooled) readiness for each emergency (ie, SRI/signal function and clinical cascade); and (4) mean overestimated readiness (pooled mean for SRI/signal function readiness minus clinical cascade readiness). Mean overall (pooled) readiness was calculated using the mean individual readiness estimate for each emergency (ie, SRI/signal function and clinical cascade). First, we reported emergency readiness estimates for all facilities. Next, we stratified emergency readiness by facility characteristics (level of care (reported C-section capability), ownership (public/government vs private) and country (Kenya vs Uganda)). Reported C-section service delivery at a facility

was used as a proxy for identifying designated CEmOC facilities (since designated BEmOC facilities do not have this capability).

For the purpose of calculating the drop-off in readiness according to the clinical cascade model, we reported the decrease in the number of percentage points in facilities ready to handle the five emergencies by stage of care of the clinical cascade, by emergency and overall across the three stages of care for five emergencies (pooled mean). To determine readiness drop-off by stage of care, we subtracted readiness (ie, percentage of facilities ready to manage the five emergencies) at the end of a stage from readiness at the end of the preceding stage. For example, to calculate readiness drop-off during stage 1 (*identify*) for each emergency, we subtracted readiness at the end of stage 1 from 100% and then calculated the mean and SD across all five emergencies by stage. This provided the pooled mean readiness drop-off across all three stages of care (*identify*, *treat*, *monitor-modify*). From these estimates, we calculated drop-off in emergency readiness by clinical cascade by calculating the mean across the three stages of care for each clinical cascade (ie, emergency) and producing five estimates of mean drop-off in readiness. The pooled mean across the five cascades determined the overall drop-off in readiness across emergencies and stages of care. Next, estimates were stratified by (1) level of care; (2) facility ownership; and (3) country.

### Patient and public involvement

The parent study results were shared with health providers at the facilities. The research question and outcome measures for this nested study of previously collected PTBi data were shaped by global maternal survival research priorities and reported experiences of women of reproductive age. For this nested study, the study findings will be shared directly with the participating PTBi facilities. Patients and the public were not directly involved in the design or conduct of the research.

## RESULTS

### Facility characteristics

In online supplemental tables S2–S3, we present facility characteristics. Of the 23 facilities, 82.6% were government-owned and 17.4% were privately owned hospitals. Thirteen of the 23 facilities did not have reported C-section capabilities (ie, were designated BEmOC facilities). All Ugandan facilities reported C-section capability and are, therefore, designed to offer CEmOC facilities. By contrast, 76.5% of the Kenyan facilities were designated BEmOC hospitals.

### Signal function estimates of emergency readiness
#### Overall readiness

The overall mean estimate of readiness as defined by the availability of tracer items for the five SRI/signal functions was 69.6% (table 1). There was little variability in readiness across the five signal functions (ranging from 60.9%

**Table 1** Emergency readiness estimates comparing clinical cascades and signal functions, all facilities*

| Clinical cascade (Signal function) | Signal functions % readiness, tracer items | Clinical cascades % readiness, stage 2 | Overestimated readiness Percentage point difference [Signal functions (–) cascade] |
|---|---|---|---|
| **Medical treatments** | | | |
| Manage sepsis—infection (Antibiotic) | 69.6 | 47.8 | 21.7 |
| Manage haemorrhage (Uterotonics) | 60.9 | 60.9 | 0.0 |
| Manage hypertensive emergency (Anticonvulsant) | 78.3 | 26.1 | 52.2 |
| Medical readiness, overall mean (pooled) | 69.6 | 44.9 | 24.6 |
| **Manual procedures** | | | |
| Manage retained placenta (Removal of retained placenta) | 69.6 | 43.5 | 26.1 |
| Manage incomplete abortion (Removal of retained products of conception) | 69.6 | 56.5 | 13.1 |
| Manual readiness, overall mean (pooled) | 69.6 | 50.0 | 19.6 |
| Overall mean readiness (pooled) | 69.6 | 47.0 | 22.6 |
| | Signal function estimate | Cascade estimate | Percentage point overestimation by signal functions |

*n=23 facilities.

for uterotonics to 78.3% for anticonvulsants). The 69.6% SRI/signal function estimates of readiness did not differ for emergencies requiring manual procedures compared with those requiring medication therapy.

### Readiness by facility characteristics
As expected, SRI/signal function readiness estimates differed by level of care (online supplemental table S4), facility ownership (online supplemental table S5) and country (online supplemental table S6). Overall, BEmOC readiness estimates were higher among facilities with C-section capability (designated CEmOC facilities), those that were privately owned and those in Uganda.

### Clinical cascade readiness
#### Readiness against the SRI/signal functions
Across all five emergencies, the signal function tracers overestimated obstetric emergency readiness by 22.6 percentage points (table 1, online supplemental figure S1). The pooled mean signal function readiness estimate was 69.6%, while readiness as measured by stage 2 of the cascade model was substantially lower at 47.0%. Notably, there was wide variability in the signal function overestimate ranging from 0.0 (manage haemorrhage/uterotonics) to 52.2 percentage points (hypertensive emergencies/anticonvulsant, table 1). The 23 percentage point overestimation seen in the full sample remained largely consistent after stratifying by facility characteristics (table 2).

### Readiness by emergency
When examining clinical cascade estimates of emergency readiness by emergency at the individual resource level, we saw an overall pattern of readiness drop-off across the stages of care (from stage 1 to stage 3, online supplemental table S7, figures S2–S6). The amount of readiness drop-off at each stage and the specific resources driving the drop-off differed by emergency.

### Readiness by cascade
There were differences in readiness drop-off along the stages of care from identification of the obstetric emergency (stage 1) through monitoring or modifying therapy (stage 3, table 3). These differences varied least for the sepsis cascade (SD=5.0) and most for the hypertension cascade (SD=35.4). There was also substantial variability in *when* the drop-off in readiness occurred. For hypertensive emergencies, the largest drop-off occurred in the identification stage (69.6%). In contrast, there was no readiness drop-off in the identification stage for the haemorrhage, retained placenta or incomplete abortion cascades. Of note, stage 1 for the haemorrhage and retained placenta cascades is based on staff skill alone. For this commodity-based study, we assumed staff skill to be 100% for all emergencies, but it was not measured (skill is explicitly shown for the haemorrhage cascade since it does not require any commodities for identification).[21]

When examining drop-off in readiness by facility characteristics, there were subtle differences by level of care (online supplemental table S8), ownership (online

**Table 2** Overestimation of emergency readiness by signal function tracers compared with clinical cascades, by facility characteristics

| Clinical cascade (Signal function) | Percentage point difference in readiness [Signal functions (–) cascade] | | | | | | |
| | Reported C-section capability | | | Ownership | | Country | |
| | **All** | **C-section** | **No C-section** | **Government** | **Private** | **Kenya** | **Uganda** |
|---|---|---|---|---|---|---|---|
| n | 23 | 10 | 13 | 19 | 4 | 17 | 6 |
| Manage sepsis—infection (Antibiotic) | 21.7 | 40.0 | 7.7 | 21.1 | 25.0 | 11.8 | 50.0 |
| Manage haemorrhage (Uterotonics) | 0.0 | 0.0 | 0.0 | 0.0 | 0.0 | 0.0 | 0.0 |
| Manage hypertensive emergency (Anticonvulsant) | 52.2 | 60.0 | 46.2 | 47.4 | 75.0 | 52.9 | 50.0 |
| Manage retained placenta (Removal of retained placenta) | 26.1 | 0.0 | 46.2 | 31.6 | 0.0 | 35.3 | 0.0 |
| Manage incomplete abortion (Removal of retained products of conception) | 13.1 | 10.0 | 15.4 | 10.5 | 0.0 | 11.8 | 16.7 |
| Mean percentage point overestimation by signal functions (pooled) | 22.6 | 22.0 | 23.1 | 22.1 | 20.0 | 22.4 | 23.3 |

C-section, caesarean section.

supplemental table S9) and country (online supplemental table S10). For instance, at facilities *with* reported C-section capability, the percentage of drop-off across the stages of care varied least for sepsis (SD=17.3) and most for haemorrhage (SD=30.0, online supplemental table S8). Among facilities *without* C-section capability, the

**Table 3** Mean drop-off in readiness by cascade and stage of care among all facilities*

| Readiness drop-off by stage of care | | | | Readiness drop-off by emergency | |
| | **1** | **2** | **3** | | |
| Clinical cascade | **Identify** | **Treat** | **Monitor-modify** | **Mean drop-off across three cascade stages of care** | **SD** |
|---|---|---|---|---|---|
| | – | – | – | 28.4%† | 24.2‡ |
| Sepsis—infection | 26.1% | 26.1% | 17.4% | 23.2% | 5.0 |
| Haemorrhage | 0.0% | 39.1% | 47.8% | 29.0% | 25.5 |
| Hypertensive emergency | 69.6% | 4.4% | 13.0% | 29.0% | 35.4 |
| Retained placenta | 0.0% | 56.5% | 34.8% | 30.4% | 28.5 |
| Incomplete abortion | 0.0% | 43.5% | 47.8% | 30.4% | 26.5 |
| **Overall drop-off by stage of care** | | | | | |
| Drop-off by stage of care (across all emergencies), pooled mean | 19.1% | 33.9% | 32.2% | | |
| SD | 30.4 | 19.8 | 16.4 | 3.0§ | |

*n=23 facilities.
†Pooled mean readiness drop-off across three clinical cascade stages of care and five emergencies.
‡Mean of the SDs.
§SD across three stages of care and five emergencies.

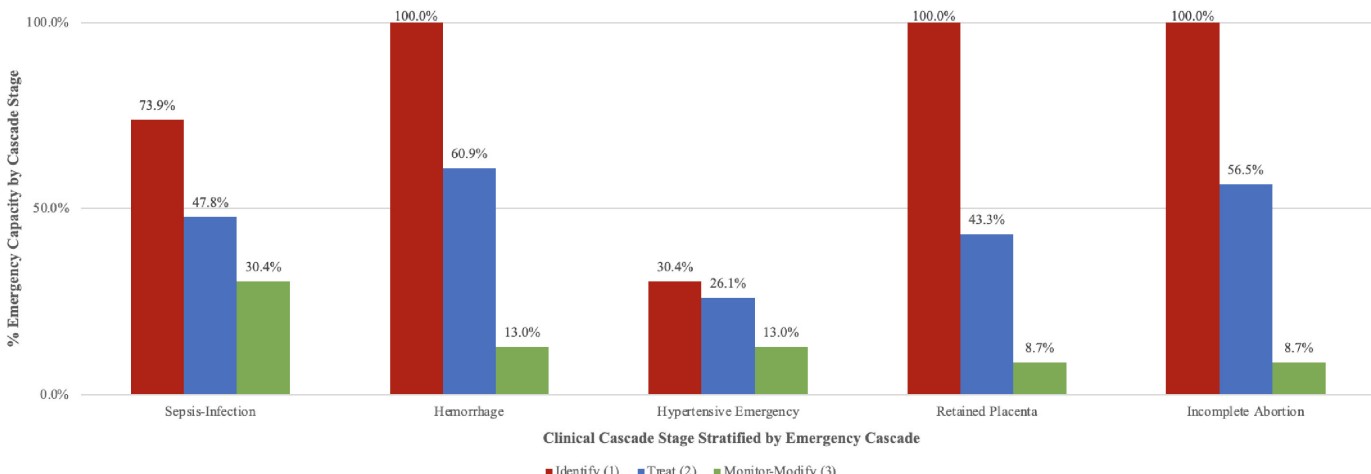

**Figure 1** Emergency readiness estimates by emergency cascade and stage of care.

drop-off pattern was the same as seen in the full sample—it varied least for sepsis (SD=11.8) and most for hypertensive emergencies (SD=42.4).

### Readiness by stage of care

This study revealed a pattern of 28.4% drop-off in readiness across emergencies and stages of care (SD=3.0) despite moderate variability in where the drop-off occurred across these stages (average SD across stages=24.2, table 3). Overall, the largest drop-off took place in *treating* the obstetric emergency (stage 2; 33.9%), and the smallest drop-off took place in identifying the emergency (stage 1; 19.1%). Figure 1 depicts the pattern of readiness drop-off across the stages of care and emergencies.

The overall readiness drop-off based on facility characteristics (reported C-section capacity, ownership and country) was similar to that observed in the full facility sample (table 4). For instance, we found similar drop-off and variability based on reported C-section capacity. There was a 26.7% drop-off in readiness across emergencies and

stages of care among facilities with C-sections (SD=4.1) and 29.7% among those without (SD=3.4). Consistent with the unstratified results, the largest drop-off in readiness among facilities without reported C-section capability occurred in the treatment stage (43.1%) (online supplemental table S8). However, among facilities with reported C-section capability, the largest drop-off occurred in the final stage of care—monitoring-modifying therapy (38.0%). Similar patterns emerged across facility ownership (table 4, online supplemental table S9) and country (table 4, online supplemental table S10).

### DISCUSSION

This study expands the global scholarship on basic obstetric emergency readiness by confirming previously published patterns of (1) resource loss across three stages of emergency care (*identify*, *treat*, *monitor-modify*) and (2) signal function tracer overestimation of emergency

**Table 4** Mean drop-off in readiness by facility characteristics

| | Mean drop-off across three cascade stages of care | | | | | | |
| | Reported C-section capacity | | | Ownership | | Country | |
| Clinical cascade | All | C-section | No C-section | Government | Private | Kenya | Uganda |
|---|---|---|---|---|---|---|---|
| n | 23 | 10 | 13 | 19 | 4 | 17 | 6 |
| Sepsis—infection (%) | 23.2 | 20.0 | 25.6 | 26.3 | 8.3 | 21.6 | 27.8 |
| Haemorrhage (%) | 29.0 | 30.0 | 28.2 | 28.1 | 33.3 | 27.5 | 33.3 |
| Hypertensive emergency (%) | 29.0 | 30.0 | 28.2 | 28.1 | 33.3 | 27.5 | 33.3 |
| Retained placenta (%) | 30.4 | 26.7 | 33.3 | 31.6 | 25.0 | 31.4 | 27.8 |
| Incomplete abortion (%) | 30.4 | 26.7 | 33.3 | 31.6 | 25.0 | 31.4 | 27.8 |
| **Overall drop-off** | | | | | | | |
| Mean* (%) | 28.4 | 26.7 | 29.7 | 29.1 | 25.0 | 27.8 | 30.0 |
| SD† | 3.0 | 4.1 | 3.4 | 2.4 | 10.2 | 4.0 | 3.0 |

*Pooled mean readiness drop-off across three stages of care and five emergencies.
†SD of mean drop-off across three stages of care and five emergencies.
C-section, caesarean section.

readiness. It expands previous scholarship by high-lighting how emergency readiness differs by level of care, facility ownership and country. Results from this study aligned with those found in the published comparison of SRI/signal functions and clinical cascades. The signal functions consistently overestimate obstetric emergency readiness. We found that signal functions overestimated readiness by 22.6 percentage points. This is about half of what was reported in the formative study (54.5 percentage points).[21] However, the formative study only included designated BEmOC facilities, whereas nearly half (43.5%) of the facilities in this study were designated CEmOC facilities. Despite some operational (eg, variable inclusion/definitions) and contextual (eg, country/region) differences between studies, any overestimation of practical emergency readiness may be profoundly clinically relevant—particularly since elevated maternal mortality persists globally despite increased rates of facility-based deliveries. The 22.6 percentage point overestimation of readiness by the signal functions in this study—particularly since it includes designated CEmOC facilities—is profoundly concerning for advancing global maternal survival.

As in previous studies, we found a consistent drop-off in readiness across the stages of care. Across three stages of care for five emergencies, we found a mean 28.4% drop-off in readiness which was profoundly consistent (SD=3.0, table 3). The published study of 44 BEmOC facilities in Kenya found a very similar drop-off pattern of 33.0% which also varied little (SD=0.4).[21] Moreover, in a study of the clinical cascades as applied to neonatal emergencies, the mean drop-off in readiness was 30.0% across emergencies and stages of care (based on 2016 data).[24] Given that the 30% aggregate drop-off for neonatal emergencies comes from the same facilities used in this maternal study, the mean readiness drop-off indicator may be an indicator of system-level emergency readiness, as suggested previously.[21] Together, these three studies suggest a drop-off in readiness of approximately 30% across emergencies and stages of care in both maternal and neonatal contexts across country contexts and levels of care. Together with the stages of care and condition-specific indicators, the aggregate drop-off may be used to guide resource allocation and interventions and as a means of monitoring and comparing readiness across facilities, health systems, countries or geographic regions. There is growing evidence that global multinational agencies, programme planners and policy makers should consider replacing the current SRI methods and the signal function estimates of readiness with the clinical cascades.

This is the first known study to measure how country, facility ownership and level of care impact clinical cascade readiness. Overall estimates of readiness (according to both signal functions and clinical cascades) were higher among Ugandan than Kenyan facilities (73.3% vs 68.2% according to the signal functions and 50.0% vs 45.9% according to the clinical cascades). These results were expected given the level of care offered at the health facilities included in each country's sample. Ugandan

facilities were largely district or regional hospitals (all CEmOC) while the Kenyan facilities were primarily smaller subcounty hospitals and health centres. Likewise, estimates of readiness (based on both signal functions and cascades) were higher among private facilities than government facilities, with nearly all private facilities prepared to handle basic obstetric emergencies according to the signal clinical cascades (75.0%). Differences in readiness by level of care (among CEmOC vs BEmOC-designated facilities) were evident as well (and, as previously mentioned, likely driving differences by facility ownership and country). By definition, all facilities with C-section capability are designated CEmOC facilities that should be prepared to manage all BEmOC emergencies in addition to performing C-sections and blood transfusions. However, this study demonstrated that even designated CEmOC facilities were not ready to perform all BEmOC functions. This gap was present when emergency readiness was estimated using both the signal functions (80.0%) and clinical cascades (58.0%). This gap suggests that facilities designated to offer CEmOC were not practically equipped to handle the complexity of obstetric emergencies required by their designation. If seen across a larger sample, the results could suggest the need for additional investment to equip designated CEmOC facilities with the supplies needed to handle the full range of obstetric emergencies.

This study has six primary limitations. First, there was a slight variation in the resources used to construct the signal functions and clinical cascades between this study and the two published studies because this study used updated WHO definitions that were not used in the first two studies. Further, there are minor between-country variations in how some commodities are defined. Second, we used data from a single point in time. Given the unpredictable availability of consumable resources and drugs in some global health systems, estimating emergency readiness using a single time point may be insufficient to capture a facility or system's long-term readiness.[18 28] Collecting data from the same set of facilities at more frequent intervals could provide more accurate readiness data and reveal patterns in resource availability based on supply procurement. Third, due to the absence of select variables, we used proxies for missing tracer items. These proxies may not capture the nuances of actual resource availability. For instance, facilities without electricity often use kerosene-powered refrigerators. With information on neither refrigeration nor kerosene power availability, we used electricity alone as a proxy for refrigeration needed to store select drugs. Consequently, facilities without electricity and using kerosene-powered refrigerators would be unintentionally reported as lacking refrigeration in these analyses. Fourth, more broadly, measuring the availability of resources alone does not take into account the quality of the resource, the number of back-up resources available for sustained care nor clinicians' ability to use the resources effectively for emergency care.[24 29] Both the signal functions/SRI and clinical cascade estimates of readiness rely on emergency supply availability. Consequently, in commodity-based

metrics, it is not possible to estimate actual clinician skill for diagnosing and treating disorders. By extension, although the clinical cascade stages of care are based on clinical action (eg, identify/diagnose, treat, monitor-modify), it is difficult to reliably and affordably measure these clinical actions at scale in national or regional health systems. When possible to measure, information on resource functionality (eg, drug expiration dates), facility staffing levels, clinician knowledge and skill, and actual reported performance of the six signal function clinical actions would allow for more precise estimates of practical emergency readiness at the facility level. Fifth, although resources were reported as present in the facilities, we could not determine how accessible resources were during emergencies. Given the importance of timing in an emergency, resource location and availability are critical and should be explored in future research. Sixth, future research could link cascade estimates of readiness to facilities' adverse clinical outcomes (eg, MMR, maternal near miss, severe maternal outcomes or prevalence of specific emergencies) to determine the empirical relationship between clinical cascade estimates of readiness and obstetric clinical outcomes.

Despite these limitations, this study offers important contributions to the emerging evidence on the use of clinical cascades as a more precise and targeted alternative to signal function estimates of emergency readiness. This conclusion adds to mounting concerns regarding the accuracy of the signal functions as an indicator of practical obstetric emergency readiness in the real world. By defining the resources necessary to identify and treat emergencies and then monitor-modify treatment as clinically indicated, the cascades provide a detailed stepwise analysis of resource availability and a novel set of readiness indicators. First, the cascades show precisely where the drop-off in readiness occurs. Second, the drop-off can be estimated at the stage of care, obstetric emergency and individual resource level. Third, there is growing preliminary evidence that the variability in drop-off across stages of care and cascades may provide an indicator of emergency readiness at the system level. Results from this study could inform strategies that optimise emergency commodity provision for Kenyan and Ugandan facilities and strengthen system-level strategies for improving emergency-specific maternal survival globally. While measuring readiness according to the clinical cascade model requires a marginal increase in effort during data collection, this approach provides a more nuanced picture of capacity to manage obstetric emergencies. Further investigation into diverse contexts (eg, additional countries, regions, levels of care) is warranted and could further support existing calls to shift away from standard readiness metrics (such as the signal functions and SRI) towards the more nuanced, precise and practical estimates offered by the clinical cascades.

## CONCLUSIONS

In conclusion, emergency care is critical in managing the obstetric complications driving elevated MMRs globally and, particularly, in sub-Saharan Africa. The prominent role emergency care plays in the obstetric experience mandates an accurate measurement of readiness to handle such complications. This study suggests the need to reconsider the signal functions as the preferred method of measuring readiness, which aligns with findings from previous studies. The clinical cascades provide a more nuanced picture of clinical care that can be used to optimise emergency commodity provision and strengthen system-level strategies for improving emergency-specific maternal survival.

**Author affiliations**
[1]Behavioral, Social and Health Education Sciences, Emory University, Atlanta, Georgia, USA
[2]Institute for Global Health Sciences, University of California San Francisco, San Francisco, CA, USA
[3]Department of Obstetrics and Gynecology, Kenyatta University, Nairobi, Kenya
[4]School of Public Health, Makerere University, Kampala, Uganda
[5]Department of Obstetrics, Gynecology and Reproductive Sciences, University of California San Francisco, San Francisco, California, USA
[6]Woodruff Health Sciences Center, Emory University, Atlanta, Georgia, USA

**Acknowledgements** We thank Damien Scogin for his graphic design and Cecilia Roach for her copy-editing contributions. We appreciate all the facility healthcare workers, leadership and administration who made this work possible. We thank all members of the Preterm Birth Initiative Kenya and Uganda Implementation Research Collaborative, particularly the data teams who helped strengthen and collect data from facility maternity registers for the parent PTBi study. We are grateful to the facility clinicians and study participants for their valuable contribution to the parent PTBi study and by extension this nested analysis. Furthermore, we thank the research teams at UCSF, Makerere University, Kenya Medical Research Institute and Emory University for their hard work, collaboration and ongoing commitment to maternal survival.

**Contributors** All authors listed are responsible for this study and have participated in the concept and design, analysis, and interpretation of data, and drafting and revising of the manuscript. DW, EB, AW and PW conceptualised, secured funding and oversaw data collection for the parent study. BW and JNC conceptualised the nested analysis. BW analysed the data and wrote the first draft of the submitted manuscript, with assistance from JNC and JMS. All authors provided feedback and commented on the manuscript. BW is responsible for the overall content as the guarantor.

**Funding** This work was supported, in whole or in part, by the Bill & Melinda Gates Foundation (Grant # OPP1107312). Bill & Melinda Gates Foundation funded primary data collection and article processing charges. Under the grant conditions of the Foundation, a Creative Commons Attribution 4.0 Generic License has already been assigned to the Author Accepted Manuscript version that might arise from this submission. Emory University Rollins School of Public Health's Global Field Experience Financial Award funded effort for the literature review.

**Competing interests** AW represents his university on the Sexual, Reproductive and Child Health Committee for the Bill & Melinda Gates Foundation. JNC has grants from UNICEF and is an advisory board member for Towards Unity in Health and chair for the Community-Based Primary Health Care working group with the American Public Health Association, International Health Section. DW is on the Board of Directors of PRONTO International, a not-for-profit agency.

**Patient and public involvement** Patients and/or the public were not involved in the design, or conduct, or reporting, or dissemination plans of this research.

**Patient consent for publication** Not required.

**Ethics approval** The Kenya Medical Research Institute (KEMRI/SERU/ CCR/0034/3251), Makerere University School of Public Health (MUSPH HDREC 395) and the UCSF (16-19162) Institutional Review Boards (IRBs) approved the trial prior to primary data collection. Emory University's IRB determined the nested analysis of these deidentified data was not human subjects research (STUDY 00001202).

**Provenance and peer review** Not commissioned; externally peer reviewed.

**ORCID iDs**
Bridget Whaley http://orcid.org/0000-0003-0312-7800
Elizabeth Butrick http://orcid.org/0000-0002-0026-7464
Jessica M Sales http://orcid.org/0000-0002-1571-8192
Anthony Wanyoro http://orcid.org/0000-0001-7167-3008
Peter Waiswa http://orcid.org/0000-0001-9868-0515
Dilys Walker http://orcid.org/0000-0002-9599-388X
John N Cranmer http://orcid.org/0000-0001-7053-9157

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
