## [Reviewer comments · BMJ Open]

ARTICLE DETAILS

TITLE (PROVISIONAL)	Using clinical cascades to measure health facilities' obstetric emergency readiness: testing the cascades model using cross-sectional facility data in East Africa
AUTHORS	Whaley, Bridget; Butrick, Elizabeth; Sales, Jessica; Wanyoro, Anthony; Waiswa, Peter; Walker, Dilys; Cranmer, John

VERSION 1 – REVIEW

REVIEWER	Bailey, Patricia E. unaffiliated - consultant
REVIEW RETURNED	26-Oct-2021

GENERAL COMMENTS	This manuscript appears to be the 3rd paper to use a clinical cascade approach to assess readiness related to maternal and newborn signal functions (the latter are not yet formally selected but currently are undergoing a process for inclusion into a revised EmONC framework). The authors' definition of readiness is operationalized by lists of tracer items, mostly drugs, supplies and equipment, including condition-specific protocols and sources of light. The authors contend that they are comparing signal function readiness with readiness defined by 3 sequential stages in the process of clinical care: 1) identification of an obstetric complication or emergency; 2) treatment of the condition; and 3) subsequent monitoring and modification of treatment. This nuanced focus on the readiness to detect, treat and monitor is likely to resonate with practicing clinicians but measurement of the last step – monitoring – seems to fall short of expectations (both in this paper and the Cranmer paper). I wonder if the authors could discuss this as a limitation to this approach and how measuring this stage of care might be improved? The comparison between readiness of the signal function (SF) and readiness as portrayed by each of the stages of care does not make a lot of sense to me since most of the items found in the list for SF readiness are also found in the treatment list. Perhaps I have missed something – is this list taken from the SRI index for SF readiness? My understanding from the authors is they are critiquing the SRI as it is based on a long list of items and a facility is subsequently given a readiness score based on the index or, if more nuanced, on different domains of the index. Nevertheless, I think the focus on the 3 steps in the care process is very helpful as it reveals where in the process the availability of key items is lacking i.e. where readiness drops off. However, to be most helpful to programs, the status of individual items must be shared.
---

The reason I find the comparison between SF and the clinical cascade readiness unsatisfactory may be rooted in my understanding of the intent of the SFs. They were selected because they map to the treatment of the obstetric complications that lead to the major direct causes of maternal death. They were conceived of as a way to define levels of care – basic EmOC and comprehensive EmOC. But the critical metric has been the performance of the SFs, which defines whether a facility functions fully at either of these 2 levels. (I suggest that the authors be careful how they call these facilities – a more accurate terminology might be “designated BEmOC or CEmOC” or “potential” BEmOC. This nomenclature of “fully functioning” and how strict one wants to be about requiring a facility to have performed all 9 of the SFs to be considered functional at the CEmOC level and all 7 (including neonatal resuscitation) for BEmOC is still under debate but most facilities currently classified as BEmOC fail to function at that level, as the authors in this study indicate in the discussion.)

The cascade approach is helpful to understanding where the commodity gaps are in terms of readiness but many other reasons exist to help explain why SFs are not performed (reasons related to human resources – too few, lacking confidence, competency deficits, etc.; national policies, facility policies; weak facility management; no patients requiring a specific SF/intervention). Performance of the SFs is ultimately what we want to know because that indicates whether the health system is functioning as expected. Readiness in terms of tracer items, electricity/power, protocols are all important but if the SFs are not being performed we have a problem. One of the assumptions about signal functions has been if performed, there must have been skilled personnel and the drugs, equipment or supplies. We know now that we can't assume this is true. In sum, I found discussion of the “SF approach” confusing as I assumed that the authors were referring to the measurement of whether the SF had been performed/administered. My recommendation would be to drop that piece of the analysis and focus on the 3 levels of care that are based on the signal functions, complete with a table that shows the drop off from “identify” to “treat” and “treat” to “monitor.” See Figure S6. I would find a place for Figure S6 in the body of the paper. To me, it is the most important table of the paper.

The stratification by level of care, ownership and country is interesting but the most important factor is clearly level of care. Readiness should differ by levels of care and it explains many of the differences between the 2 countries and between mission vs government facilities (I suspect the few mission facilities are hospitals with CEmOC potential).

In the methods, please clarify how you analyze the 3 stages of care. I am assuming that when you analyze stage 2 (treatment), you also take into consideration all the items listed under stage 1 (identification), and thus stage 2 by definition has to be equal to or smaller than stage 1.

The paper could use a strong editor to tighten up the text. Some of the findings are not of programmatic interest (when no clear pattern emerges). Some of the tables could be dropped.

Please see the notes on the manuscript itself for additional observations.

REVIEWER	Koigi, Paul Nairobi Hospital, Obstetrics/Gynecology
REVIEW RETURNED	09-Nov-2021

GENERAL COMMENTS	This article has a lot of value to offer in the East African and international contexts. Emergency preparedness is perhaps the single biggest factor that determines if a mother survives untoward obstetric complications, perhaps second only to the patient's decision to actually seek skilled medical attention. The insights offered here would have even higher value if this study was replicated in various settings to enable cross-cutting comparisons. Overall, this is a very well designed and executed study that has made previously unquantified aspects of care become more objectively measurable, thereby providing room for improvement at both individual facility and systemic levels.
---

REVIEWER	Opondo, Charles London School of Hygiene & Tropical Medicine, Department of Medical Statistics
REVIEW RETURNED	15-Nov-2021

GENERAL COMMENTS	This study is well conducted and reported. I have a some comments pertaining to the methods and results. In the statistical analysis, given that you are estimating percentages for inference, it would be useful to also report confidence intervals for these estimates, not standard deviations which are generally descriptive and, for proportions/percentages are a function of the proportion therefore not terribly informative. You should also be clearer that what you are reporting as 'means' are actually pooled proportions/percentages. Further in the statistical analysis, it is unclear what hypotheses are being tested by the Fisher's exact tests; you simply say that you planned "to compare readiness estimates for significance" but it is not clear what comparison was being made or why it was necessary to make them. The results are a bit clearer on what the comparisons were, but it is still unclear how they fit within the objectives of the study. The descriptive summary of the sample should generally not be relegated to supplementary tables. Where you indicate that signal function estimates differed by some group (e.g. pg 7 line 3-12, Table 1), it would have been more informative to report the estimates across the groups with the 95% confidence intervals and/or some formal test to evaluate evidence of a difference.
---

VERSION 1 – AUTHOR RESPONSE

Reviewer	Comment	Response
Bailey	This manuscript appears to be the 3rd paper to use a clinical cascade approach to assess readiness related to maternal and newborn signal functions (the latter are not yet formally selected but currently are undergoing a process for inclusion into a revised EmONC framework).	Thank you for your comment.
Bailey	The authors' definition of readiness is operationalized by lists of tracer items, mostly drugs, supplies and equipment, including condition-specific protocols and sources of light. The authors contend that they are comparing signal function readiness with readiness defined by 3 sequential stages in the process of clinical care: 1) identification of an obstetric complication or emergency; 2) treatment of the condition; and 3) subsequent monitoring and modification of treatment. This nuanced focus on the readiness to detect, treat and monitor is likely to resonate with practicing clinicians but measurement of the last step – monitoring – seems to fall short of expectations (both in this paper and the Cranmer paper). I wonder if the authors could discuss this as a limitation to this approach and how measuring this stage of care might be improved?	We added information to the Discussion section to address this comment on the limitations of measurement.
Bailey	The comparison between readiness of the signal function (SF) and readiness as portrayed by each of the stages of care does not make a lot of sense to me since most of the items found in the list for SF readiness are also found in the treatment list. Perhaps I have missed something – is this list taken from the SRI index for SF readiness? My understanding from the authors is they are critiquing the SRI as it is based on a long list of items and a facility is subsequently given a readiness score based on the index or, if more nuanced, on different domains of the index. Nevertheless, I think the focus on the 3 steps in the care process is very helpful as it reveals where in the process the availability of key items is lacking i.e. where readiness	Thank you for your comment. We point the readers to the supplemental files for the individual resource/emergency-level data. We have these data and would eagerly submit them for consideration with this manuscript. However, we were limited in the number of tables we could present by author guidelines. If the reviewer-editor team would permit us to include this information, we would be delighted to share it as a table in the body of the manuscript.

	drops off. However, to be most helpful to programs, the status of individual items must be shared.	
Bailey	The reason I find the comparison between SF and the clinical cascade readiness unsatisfactory may be rooted in my understanding of the intent of the SFs. They were selected because they map to the treatment of the obstetric complications that lead to the major direct causes of maternal death. They were conceived of as a way to define levels of care – basic EmOC and comprehensive EmOC. But the critical metric has been the performance of the SFs, which defines whether a facility functions fully at either of these 2 levels. (I suggest that the authors be careful how they call these facilities – a more accurate terminology might be “designated BEmOC or CEmOC” or “potential” BEmOC. This nomenclature of “fully functioning” and how strict one wants to be about requiring a facility to have performed all 9 of the SFs to be considered functional at the CEmOC level and all 7 (including neonatal resuscitation) for BEmOC is still under debate but most facilities currently classified as BEmOC fail to function at that level, as the authors in this study indicate in the discussion.)	We added "designated" in front of "BEmOC facilities" and "CEmOC facilities" and added clarification on why we do not measure the performance of the signal functions in our study. Further, we are adapting the SRI % indicators which are based on the signal functions. Consequently, based on this helpful feedback, we have added SRI/Signal Functions descriptions to make more explicit that we are borrowing from SRI's % readiness indicators.
Bailey	The cascade approach is helpful to understanding where the commodity gaps are in terms of readiness but many other reasons exist to help explain why SFs are not performed (reasons related to human resources – too few, lacking confidence, competency deficits, etc.; national policies, facility policies; weak facility management; no patients requiring a specific SF/intervention). Performance of the SFs is ultimately what we want to know because that indicates whether the health system is functioning as expected. Readiness in terms of tracer items, electricity/power, protocols are all important but if the SFs are not being performed we have a problem. One of the assumptions about signal functions has been if performed, there must have been skilled personnel and the drugs, equipment or supplies. We know now that we can't assume this is true. In sum, I found discussion of the “SF approach” confusing as I assumed that the authors were referring to the measurement of whether the	Thank you for this thoughtful reflection on the nuanced relationship between reported signal function performance and the SRI indicators of % readiness for obstetric emergencies based on SRI. We added Figure S6 to the body of the article as Figure 1. We replaced "Signal Functions" with "SRI/Signal Functions" to increase clarity.

	SF had been performed/administered. My recommendation would be to drop that piece of the analysis and focus on the 3 levels of care that are based on the signal functions, complete with a table that shows the drop off from "identify" to "treat" and "treat" to "monitor." See Figure S6. I would find a place for Figure S6 in the body of the paper. To me, it is the most important table of the paper.	
Bailey	The stratification by level of care, ownership and country is interesting but the most important factor is clearly level of care. Readiness should differ by levels of care and it explains many of the differences between the 2 countries and between mission vs government facilities (I suspect the few mission facilities are hospitals with CEmOC potential).	We re-focused our manuscript on the level of care analysis, with detailed information on the other two analyses in the supplemental files.
Bailey	In the methods, please clarify how you analyze the 3 stages of care. I am assuming that when you analyze stage 2 (treatment), you also take into consideration all the items listed under stage 1 (identification), and thus stage 2 by definition has to be equal to or smaller than stage 1.	We added language in the "Clinical Cascades" section of the Methods to clarify this point.
Bailey	The paper could use a strong editor to tighten up the text. Some of the findings are not of programmatic interest (when no clear pattern emerges). Some of the tables could be dropped.	Thank you for the suggestion. We reviewed and edited the overall manuscript for message cohesion. Please find extensive editorial revisions in the "track changes" manuscript. We also removed the details about differences by country and ownership from the manuscript to more precisely focus on designated CEmOC facilities.
Bailey	"the signal functions overestimated" -- are they referring to performance of SFs? what do they mean by this?	We added language to the Strengths and Weaknesses to clarify that performance of SFs was not used to define readiness in this study (as this was not measured in the parent study and the tracer item approach is consistent with the SRI methodology). Rather, we adapted the SRI % readiness indicator which is based on the signal functions model of reported service delivery.
Bailey	I believe the authors mean to say "not all designated CEmOC facilities.." If they don't perform the signal functions we shouldn't be calling the facility a CEmOC facility - it is perhaps a prioritized, designated or potential CEmOC facility.	We added "designated" in front of "CEmOC facilities"

Bailey	perhaps you should define what these 3 stages before you refer to them.	We added language to the abstract to define the three stages up front.
Bailey	not clear - resources lost.	We adjusted the wording in this sentence to clarify.
Bailey	["losses" highlighted]	We adjusted the wording in this sentence to clarify.
Bailey	Awkward (re. "Only measuring the resource availability does not provide a complete picture of emergency readiness as the clinicians' ability to use the resources effectively is critical but difficult to measure")	We adjusted the wording in this sentence to clarify.
Bailey	Some might say that SFs are a method for assessing how well a facility functions; facility and health system capability to save women's lives during and after pregnancy. I don't see them as assessing readiness - I do see how one might want to look at SF readiness, however.	Thank you for this comment, we more explicitly referred to the SRI methods which use signal function tracer items to measure readiness.
Bailey	["cascading loss of emergency obstetric resources" highlighted]	We were unsure why this phrase was highlighted. We ask that you please clarify if you are suggesting or requesting a change.
Bailey	["functions to assess readiness" highlighted]	We were unsure why this phrase was highlighted. We ask that you please clarify if you are suggesting or requesting a change.
Bailey	["accurately" highlighted]	We were unsure why this phrase was highlighted. We ask that you please clarify if you are suggesting or requesting a change.
Bailey	they might start by defining what a signal function is and how they are used, which is to say to assess whether a facility is fully functioning as B or C-EmOC IF all SFs had been performed. Performance of the SFs is the critical function. Readiness however is important and knowing this tells us something about why SFs are not performed and why facilities designated to function as B or C-EmONC fail to live up to their designation.	We addressed this comment with additional information on the SRI methodology and the justification for not measuring performance of the signal functions in this paper. This is a commodity-based study.
Bailey	Traditionally i would say that SF readiness is assessed with tracer items, or those items considered the most essential, for each SF (see EmONC assessment reports or Ghana paper).	We added this language to the Introduction, where we first define tracer items.
Bailey	This concept of CC needs more explanation of how it works, what it means, how it's applied.	We added clarification in the Introduction and Methods section.

Bailey	the sentence up to here doesn't make a lot of sense to me. I get the feeling that the authors aren't defining SFs like we tend to do.	We added clarification in the Introduction and Methods section.
Bailey	What about stage 3? you refer to stage 3 later on in the paper.	We added language to clarify in which analyses Stage 3 readiness is included.
Bailey	What is "the indiv resource level"? this means at the individual tracer item.	We added language to clarify.
Bailey	again, odd phrasing - unable to measure readiness using the signal functions what do they mean?	Thank you for this input. We added additional clarifying language.
Bailey	["comparing the clinical cascades and signal functions" highlighted]	We were unsure why this phrase was highlighted. We ask that you please clarify if you are suggesting or requesting a change.
Bailey	["overestimated readiness (signal function means readiness minus clinical cascade mean readiness)" highlighted]	We were unsure why this phrase was highlighted. We ask that you please clarify if you are suggesting or requesting a change.
Bailey	Is this programmatically of interest? i would much prefer to look at each separate SF which will tell you what you need to do.	We use overall readiness to compare the signal function and clinical cascade estimates - a focus of our study. However, we also include individual estimates for the clinical cascades in the supplemental files. If the journal allows us to add additional figures to the body of the manuscript, we can pull an example of the individual estimates from the supplemental files into the body.
Bailey	I don't think you calculated the percentage decrease - you show the decrease in number of percentage points. This is different from a percentage decrease.	You are correct in that we report decrease in percentage points, not percent decrease. We updated the language.
Bailey	I don't think there is a table that shows the readiness scores across the stages of care so that one can see how those scores drop off. The tables show instead the losses (the subtraction is has already been done).	We moved Figure S6, which shows these estimates, into the body of the article as Figure 1.
Bailey	I am not sure that because this study was nested in another that necessarily means that patients or the public were not involved.	We dropped "as this was a nested analysis of previously collected facility-level data."
Bailey	Research question in this study or the parent study?	We combined this sentence with the previous sentence to clarify that the research question we are referring to is that of the parent study.
Bailey	this almost sounds like a non sequitur (last sentence).	We removed this sentence
Bailey	Suggest not to use 2 decimal places; if you must use one, okay. Your n's are small and this level of precision not needed. Can't you	We switched "facility demographics" to "facility characteristics" throughout the article and supplemental files; We reported

	just call these facility characteristics? Do facilities have demographic characteristics - people do but ...	only 1 decimal place throughout the manuscript and tables. However, the graphics were designed by a graphic artist and contain two decimal places. We can easily edit those down to 1 decimal place as well if requested by the editor. However, we were not able to get those revised in time for this response (we are happy to have them edited).
Bailey	This is an unfortunate misunderstanding of the EmOC framework. Just because a HF offers surgery does not make it a fully functioning CEmOC facility. Please refer to these as potential or designated/prioritized as CEmOC facilities.	We switched to "designated CEmOC" or "designated BEmOC" facilities throughout
Bailey	The overall mean estimate of readiness as defined by the availability of tracer items for the 5 SFs was 69.57%.	We adopted this language.
Bailey	Can you explain the purpose of separating the 5 SFs into these 2 groups? It is true they can be divided in this way but is there a theoretical reason / added value to making this distinction? Especially in light of the fact that you are excluding AVD, the other manual procedure and missing maternal SF?	We retained this distinction because it adds a simple, intuitive sub-division which is well-established in the literature and reflects theoretical differences in emergencies addressed primarily through medications vs. those addressed primarily through procedures.
Bailey	Please discuss the rest of Table 1 before you launch into the findings found in Table S4.	Thank you for this feedback. We have chosen to keep the results structured as is in order to follow a thematic presentation of the findings that is consistent with the other relevant publications.
Bailey	Bottom line seems to be that levels of care (CS provided vs not provided) drives these differences. Uganda has only 6 facilities but they all function at a high level unlike the Kenyan HFs. Mission HFs are hospitals (with CS capes). The differences between sector and country are explained by CS capes. If showing differences between countries is critical to the study aims or conclusions, i would advise a stratified analysis by country and then by level of care (CS capes or not). That way you can compare Kenyan HFs with CS capes to Ugandan HFs with CS capes. It will also mean no comparison group for kenyan HFs with no CS capes.	We re-focused our manuscript on the level of care analysis, with detailed information on the other two analyses in the supplemental files.
Bailey	It is unfortunate that one has to go to a supplementary table to actually see what is included in the readiness scores for the SF and for the stages of care. The value of showing the tracer items for the SFs is a bit	Unfortunately, this table is too long to be placed in the body of the manuscript, per journal guidelines.

	lost to me since the list of items follows / is so aligned with stage two (treatment).	
Bailey	Why are you jumping to stage 2? Why not show stage 1, 2 and 3 all together? Has stage 1 been incorporated in order to arrive at stage 2?	Stage 2 is inclusive of Stage 1. Stage 3 is not used when comparing readiness estimates across SF and CC, given that SF does not have a quality indicators for monitoring and modifying the primary emergency treatment.
Bailey	this should be labeled "percentage point overestimation"	You are correct in that we report decrease in percentage points, not percent decrease. We updated the language.
Bailey	I think this should be called something other than SF estimate -- why not call it SF hardware tracers?	Thank you for your feedback. We retained this language to remain consistent with the previous publications on this topic/methodology.
Bailey	by 22.6 percentage points. Not percentage.	You are correct in that we report decrease in percentage points, not percent decrease. We updated the language.
Bailey	What happened at stage 1 level?	Per our previous edit, Stage 2 is inclusive of Stage 1. Stage 3 is not used when comparing readiness estimates across SF and CC, given that SF does not have a quality indicators for monitoring and modifying the primary emergency treatment.
Bailey	Not sure this adds anything at all -- if the overall sample in Table 1 gives you 22.6% you would expect each stratified analysis to do the same. Not sure you need to show this table ????????	We re-focused our manuscript on the level of care analysis, with detailed information on the other two analyses in the supplemental files.
Bailey	would condition or obstetric complication be better than disorder?	We changed the language in the article to consistently refer to obstetric emergencies.
Bailey	Be consistent with your terms - use obstetric complication or condition or emergency condition.	We changed the language in the article to consistently refer to obstetric emergencies.
Bailey	I think a table in the text and not as a supplement should show all stages of care.	Thank you for your comment. We point the readers to the supplemental files for the stages of care. We have these data and would eagerly submit them for consideration with this manuscript. However, we were limited in the number of tables we could present by author guidelines. If the reviewer-editor team would permit us to include this information, we would be delighted to share it as a table in the body of the manuscript.
Bailey	Use terminology of 2009 Handbook - uterotronics.	We updated the language to uterotronics throughout.

Bailey	This seems to be a big flaw in this study/methodology. If the study is commodity-based, then fine - don't include provider skill as an item anywhere in your stages of care and definition of readiness. However, commodities don't help in a vacuum and skilled providers must be there to identify and treat complications.	Thank you for this comment. We added clarification in the Results and Discussion section to address this.
Bailey	Shouldn't this go below on the row for Overall Drop off by Stage and re-label the category labels (to eliminate "across stage"?)	Yes, we made this change. Thank you.
Bailey	How can you call this a consistent pattern of 28.41% when it is the mean across the 5 emergencies (with a range from a low of 23 to 30??)	We switched from "consistent" to "overall"
Bailey	Refer to "stages of care" not simply "stage"	We changed the language in the article to consistently refer to "stages of care."
Bailey	I worry that the aggregation across complications is not very helpful -- it certainly masks critical information that is needed for making programmatic corrections. The fact that readiness is better for some signal functions than others is important. I find the emphasis on loss of readiness a distraction. For QI sake, stressing readiness (and being more positive) is intuitive. We are not used to interpreting 0% as something positive, i.e. no loss in readiness.	Thank you for this insightful suggestion. We designed this research to critique the dominant model used to assess a facility's readiness to manage basic obstetric emergencies. Consequently, we believe it is critical to first establish how the current SRI/signal functions model overestimates practical readiness. However, we absolutely agree that practically it is most critical to show how readiness is lost across each emergency. Based on author guideline limits on the number of figures/tables, we have presented this information in the supplemental figures at present. If permitted by the editor-reviewer team to include this additional table, we would gladly present these data in a summary table.
Bailey	Table 4 has 26.67%	Thank you for catching that error. We changed it to 26.67 in the text.
Bailey	Again, don't use consistent; use "Overall" as in the table.	We switched from "consistent" to "overall"
Bailey	Don't place as footnote; create a row in the table above the complications and place the n's in that row.	We made this change for the tables that have more than 2 different n's.
Bailey	Can't have 2 footnotes #7	We adjusted the footnote numbering.
Bailey	["By definition, all facilities with C Section capability are CEmOC facilities that should be prepared to manage all BEmOC signal functions in addition to performing C-sections and blood transfusions." highlighted]	We switched to "designated CEmOC" or "designated BEmOC" facilities throughout

Bailey	long-term or stable readiness	We changed this to long-term.
Bailey	How about linking readiness to whether anyone actually performs the SFs?	We added this as a limitation in "Strengths and Weaknesses" and "Discussion"
Bailey	why do some obstetric emergencies include Staff skill while others do not? Seems to me that all of the emergencies require staff skill. Why do some objects appear under stage 2 but do not appear under tracer items, such as IV pole? Under stage 3 Modify, don't you need the antibiotics? Perhaps how this table was fleshed out may need some explanation.	Thank you for raising this important clarifying point. In the revised manuscript, we endeavored to more explicitly demonstrate the cascading relationship between the stages. Consequently, an item needed in Stage 1 is reported there but not if it is needed in subsequent stages. This is visually represented in the supplemental figures by emergency.
Bailey	So only PPH is considered here? The original EmOC framework and SF included antepartum hemorrhage.	Thank you for identifying this issue as our terminology was imprecise. We have included a clinical cascade for managing hemorrhage in general. This includes antepartum and postpartum hemorrhage. The language in the manuscript has been updated to hemorrhage.
Bailey	What does Model mean? just say ownership or managing authority or sector. Is Mission considered private not-for-profit? are there no private for profit facilities in the mix?	Removed the row in the table that said "model," as it was redundant of the table title. Changed "mission" to "private" throughout the manuscript for clarity.
Bailey	Use as column headings CS capability No CS capability or whatever you've used in earlier tables - do not use 'Yes'/'No'	We changed the language in the tables to be consistent.
Bailey	this methodology based on means and means of means (overall means) assumes that each stage of treatment is equal -- that strikes me as a bit mechanical and removed from actual practice of medicine. This feels very much like an exercise in crunching numbers rather than thoughtful analysis about the content.	Thank you for this comment. The goal of this study is to build an evidence base for simple, intuitive metrics, easily calculated to guide prioritization and action. Consequently, we propose the pooled mean loss in readiness across 3 stages for all emergencies may provide a health system indicator of overall emergency readiness. This pooled mean loss in this maternal study was similar to the pooled mean loss for the newborn emergency study from these same studies--suggesting this proposed system-level indicator may be relevant. Further, we adapted the SRI/signal function methods for estimating % emergency readiness--but expanded this SRI approach to provide 1) a % readiness for each emergency and 2) more precise estimates of where readiness is lost (from identification to treatment of the emergency). Also, these indicators are based on observations of available resources in facilities that can be rapidly conducted. Although this method has

		limitations, we suggest is may strike a balance of practical/clinical relevance while being much quicker to measure compared to more resource intensive approaches that measure knowledge, skill or space-time service delivery.
Koigi	This article has a lot of value to offer in the East African and international contexts. Emergency preparedness is perhaps the single biggest factor that determines if a mother survives untoward obstetric complications, perhaps second only to the patient's decision to actually seek skilled medical attention. The insights offered here would have even higher value if this study was replicated in various settings to enable cross-cutting comparisons. Overall, this is a very well designed and executed study that has made previously unquantified aspects of care become more objectively measurable, thereby providing room for improvement at both individual facility and systemic levels.	Thank you for your feedback. As mentioned in the Discussion, we agree that replication in other contexts would offer further validation of our results. Our team will continue to expand and apply this important work to additional settings. Data collection from Ethiopia is completed and analysis nearly finalized.
Opondo	This study is well conducted and reported. I have a some comments pertaining to the methods and results.	Thank you for your feedback
Opondo	In the statistical analysis, given that you are estimating percentages for inference, it would be useful to also report confidence intervals for these estimates, not standard deviations which are generally descriptive and, for proportions/percentages are a function of the proportion therefore not terribly informative. You should also be clearer that what you are reporting as 'means' are actually pooled proportions/percentages.	Thank you for raising this terminology issue. We changed tables and text to more clearly reflect the pooled mean when we were calculating the mean of means (e.g. overall pooled mean across three stages or overall pooled mean across five emergencies).
Opondo	Further in the statistical analysis, it is unclear what hypotheses are being tested by the Fisher's exact tests; you simply say that you planned "to compare readiness estimates for significance" but it is not clear what comparison was being made or why it was necessary to make them. The results are a bit clearer on what the comparisons were, but it is still unclear how they fit within the objectives of the study.	Thank you for this clarifying insight. We removed the Fisher's exact comparison and maintained the focus on descriptive resource availability by facility characteristics.
Opondo	The descriptive summary of the sample should generally not be relegated to supplementary tables.	Thank you for this insightful comment. If the editorial team allows additional tables in the body of the manuscript, we would be happy to include this table. However, since the other Tables and Figures are critical to

		the primary message of the manuscript, they were retained.
Opondo	Where you indicate that signal function estimates differed by some group (e.g. pg 7 line 3-12, Table 1), it would have been more informative to report the estimates across the groups with the 95% confidence intervals and/or some formal test to evaluate evidence of a difference.	Thank you for raising this issue of statistical inference. At present, the manuscript is primarily designed to summarize the overall performance of the maternal clinical cascades (by emergency, by stage of care, overall) in a new context. Consequently, it is primarily designed to show overall trends and possible differences in the cascade resource availability by facility characteristics. It is beyond the scope of the current study to estimate formal statistical differences given the sample size.